# Association between plasma xanthine oxidoreductase activity and in-hospital outcomes in patients with stable coronary artery disease after percutaneous coronary intervention

**Ryota Sato[1], Keitaro Akita[1], Takenori Ikoma[1], Keisuke Iguchi[1], Takayo Murase[2], Takashi Nakamura[2], Seigo Akari[2], Satoshi Mogi[1], Yoshihisa Naruse[1], Hayato Ohtani[1], Yuichiro Maekawa[1] ***

1 Division of Cardiology, Internal Medicine III, Hamamatsu University School of Medicine, Hamamatsu, Japan, 2 Sanwa Kagaku Kenkyusho Co., Ltd., Aichi, Japan

* ymaekawa@hama-med.ac.jp

**Data Availability Statement:** All relevant data are within the paper and its Supporting Information files.

## Abstract

### Objectives

Reactive oxygen species generated by xanthine oxidoreductase (XOR) are associated with the progression of atherosclerosis. However, changes in plasma XOR (pXOR) activity after percutaneous coronary intervention (PCI) for stable coronary artery disease (CAD) remains unknown.

### Methods

Herein, we compared the change in the pXOR activity in patients undergoing PCI with that in patients undergoing coronary angiography (CAG) and further evaluated the relation between changes in pXOR activity and in-hospital and long-term outcomes of patients undergoing PCI. The pXOR activity of 80 consecutive patients who underwent PCI and 25 patients who underwent CAG during the hospitalization was analyzed daily. The percentage changes from baseline regulated time interval was evaluated.

### Results

We found that although pXOR activity decreased after PCI, and remained low until discharge, no significant changes were observed in patients undergoing CAG. Furthermore, among the patients undergoing PCI, those who experienced in-hospital adverse events, had a higher percentage of pXOR reduction 3 days after PCI. There was no association between these changes and long-term events.

**Funding:** This study was conducted as a collaborative research between Hamamatsu University School of Medicine and Sanwa Kagaku Kenkyusho Co., Ltd. Sanwa Kagaku Kenkyusho Co., Ltd. did not fund this study, but provided support in the form of salaries for TM, TN, and SA. The specific roles of these authors are articulated in the "authors' contributions section". The funder had no role in study design, data collection and analysis, decision to publish, or preparation of the manuscript.

**Competing interests:** TM, TN, and SA belong to the Sanwa Kagaku Kenkyusho Co., Ltd., which is a company marketing uric acid lowering drugs. The funder provided support in the form of salaries for TM, TN, and SA, but had no role in the study design, data collection and analysis, decision to publish, or preparation of the manuscript. This does not alter our adherence to PLOS ONE policies on sharing data and materials. The specific roles of these authors are articulated in the "authors' contributions section".

## Conclusions

A significant change in pXOR activity was observed in patients undergoing PCI than in patients undergoing CAG, and there seems to be a correlation between the in-hospital outcomes and the percentage reduction from baseline in pXOR activity.

## 1. Introduction

Oxidative stress is one of the risk factors of coronary artery disease (CAD), which affects the prognosis and reduces the survival time and quality of life of patients [1–3]. Characterized by an imbalance between the generation of reactive oxygen species (ROS) and the capacity of the intrinsic antioxidant defense system, it has been implicated in the pathogenesis of cardiovascular diseases [3]. Increased activity of xanthine oxidoreductase (XOR) is known to reportedly increase the production of both ROS and uric acid [4]. An increase in ROS production, results in oxidative stress-mediated myocardial and arterial vessel injury [5]. A technique for measuring plasma XOR (pXOR) activity was recently established, and several reports have been published regarding pXOR activity in patients with chronic heart failure and heart and renal diseases [6, 7] as well as in healthy populations [8, 9]. Although the association between single assessments of pXOR activity and cardiovascular diseases and metabolic disorders has been elucidated, changes in pXOR activity, obtained by repeated measurements, after percutaneous coronary intervention (PCI) for stable CAD, remains unknown. Therefore, we aimed to evaluate the daily pXOR activity in hospitalized patients with CAD before and after PCI and investigate the association between the changes in pXOR activity and the in-hospital and long-term outcomes.

## 2. Materials and methods

### 2.1 Study design

This study was a prospective single-center study. The study protocol complied with the guidelines of the Institutional Ethics Committee, and all the participants provided written, informed consent for participation in the study, as well as publication of their data.

### 2.2 Participants

Participants were recruited from Hamamatsu University hospital between October 2017 and December 2020. Elective coronary angiography (CAG) was performed in patients with typical angina or suspected stable CAD whose clinical characteristics and results of noninvasive testing or significant stenosis observed on coronary computed tomography indicated a high likelihood of CAD. The exclusion criteria of CAG were as follows: 1) acute coronary syndrome, 2) receiving hemodialysis, 3) post coronary artery bypass graft or 4) left ventricular ejection fraction <50%. The indications for PCI in this study were as follows: 1) Patients in whom the perfusion zone of the coronary artery with significant stenosis (American Heart Association classification >75%) coincided with the area of ischemia demonstrated by noninvasive ischemia evaluation test or 2) Patients with moderate stenosis (50–75% stenosis) or multi-vessel lesions demonstrated by CAG and ischemia by fractional flow reserve (FFR), but with a vessel diameter that allows drug-eluting stent implantation. FFR evaluation was performed following procedures previously described [10]. The exclusion criteria of PCI included acute coronary syndrome, receiving hemodialysis or PCI failure. A total of 105 patients were enrolled of

which 25 underwent elective CAG and 80 underwent PCI. Plasma XOR activity was measured daily in all the 105 subjects undergoing CAG or PCI over 3 days of hospitalization. All patients undergoing CAG or PCI were hospitalized for 3 nights unless more hospitalization days were required at the physician's discretion. All patients had a follow-up evaluation at a clinical visit at 8 months. The percentage change in pXOR activity was defined as:

% change = {(pXOR activity after CAG or PCI–pXOR activity before CAG or PCI)/pXOR activity before CAG or PCI}×100, based on the time course of the plasma XOR activity during the hospitalization.

## 2.3 CAG and PCI procedures

We used a previously described procedure for PCI [11]. Patients were administered 100 mg aspirin and 75 mg clopidogrel or 3.75 mg prasugrel orally before the scheduled PCI. CAG was performed in all patients using a 5F catheter via a radial approach [12]. Conventional methods involving the use of a guiding catheter, a 0.014-inch guidewire, and a monorail balloon catheter were used for stent deployment. The type of drug-eluting stents used was at the operator's discretion. After stent implantation, further balloon upsizing and/or higher inflation pressures were used. Platelet glycoprotein IIb/IIIa inhibitors were not used in this study.

## 2.4 Clinical assessments

Baseline characteristics of all patients were recorded. The presence of coronary risk factors, including smoking habits, hypertension, diabetes mellitus (DM), chronic kidney disease (CKD), defined as estimated GFR (eGFR) <60 mL/min/1.73 m$^2$, a family history of premature CAD, defined as myocardial infarction (MI) or sudden death in a first relative, male younger than 55 years or female younger than 65 years; and concomitant medications including aspirin, clopidogrel, prasugrel, beta-blockers, angiotensin-converting enzyme inhibitors, angiotensin II receptor blockers, calcium antagonists, and XOR inhibitors, were reviewed [11]. Further, we investigated the incidence of in-hospital and long-term outcomes including all-cause death, periprocedural MI, cardiogenic shock, stroke, a new requirement for hemodialysis, a need for blood transfusion, and major bleeding. Periprocedural MI was defined according to an expert consensus document from the Society for Cardiovascular Angiography and Interventions [13]. Major bleeding was defined as the occurrence of Bleeding Academic Research Consortium type 3 or 5 bleeding [14]. A composite in-hospital adverse event was defined as the occurrence of all-cause death, periprocedural MI, stroke, and major bleeding.

## 2.5 Laboratory methods

**2.5.1 Plasma XOR activity.** Blood samples were collected in a blood collection tube coated with ethylenediaminetetraacetic acid (EDTA)-2K kept at 4˚C until centrifugation and centrifuged by 2,000 g at 4˚C for 10 min to separate plasma. The pXOR activity was measured by liquid chromatography/triple quadrupole mass spectrometry (LC/TQMS) (Nexera HLC, SHIMADZU, Japan/QTRAP 4500, SCIEX) to detect [$^{13}C_2$, $^{15}N_2$] uric acid using [$^{13}C_2$, $^{15}N_2$] xanthine as a substrate as previously reported [15–17].

**2.5.2 Concentrations of hypoxanthine and xanthine.** Blood samples were collected using a blood collection tube coated with PAX-gene DNA, kept at 4˚C until centrifugation and centrifuged at 2,000 g at 4˚C for 10 min to separate plasma. Plasma concentrations of hypoxanthine and xanthine were measured as previously reported [18]. In brief, plasma samples were added into methanol containing [$^{13}C_2$, $^{15}N_2$] xanthine and [$^{13}C_3$, $^{15}N_3$] hypoxanthine as internal standard and were centrifuged at 3,000 g at 4˚C for 15 min. The supernatant (40 μL) was

diluted with 160 μL of distilled water and concentrations of hypoxanthine and xanthine measured using LC/TQMS.

**2.5.3 Statistical analysis.**   Continuous data including pXOR activity, which are not normally distributed are presented as median and lower upper quartile. The other continuous variables are expressed as mean ± SD if they are normally distributed. The normality of distribution of each variable was tested using the Shapiro-Wilk W test. Categorical variables are expressed as absolute values and percentages. For comparisons between CAG and PCI groups or between patients undergoing PCI with and without adverse events, the Mann–Whitney U-test (non-parametric), t-test (parametric) and Fisher's exact test or chi-square test (categorical) were used as appropriate. Within-group comparisons of a percent reduction in pXOR activity from baseline at different time points were performed using the Mann–Whitney U-test. The Wilcoxon rank-sum test was used to evaluate the changes in pXOR activity between before and 3 days after PCI. The G Power computer program version 3.1.9.2 (Heinrich Heine University, Dusseldorf, Germany) [19] was used to calculate a priori sample size required for the Wilcoxon rank-sum test. It would be modeled at an effect size of 0.5 (medium), α level of 0.05 and power of 0.80, and a minimum of 50 participants would be required. All *P* values were 2-sided, and results were considered statistically significant at a *P* value <0.05. All statistical analyses were performed using EZR (Saitama Medical Center, Jichi Medical University, Saitama, Japan), which is a graphical user interface for R (The R Foundation for Statistical Computing, Vienna, Austria, version 2.13.0). More precisely, it is a modified version of R commander (version 1.27) that was designed to add statistical functions frequently used in biostatistics.

# 3. Results

Our study population included 105 patients with CAD, of which 25 had undergone CAG and 80 had undergone PCI. Baseline characteristics of both groups of patients are shown in Table 1. The age, proportion of male, and BMI were similar between the CAG and PCI groups. There were no significant differences in the prevalence of the coronary risk factors including hypertension, DM, dyslipidemia, and smoking between the both groups. Low density lipoprotein-cholesterol levels were lower in patients undergoing PCI than in patients undergoing CAG. There were no significant differences in pXOR activity as well as in the levels of uric acid, xanthine, and hypoxanthine between the groups. As for the medication, the use of aspirin and prasugrel was higher in the PCI group than in the CAG group.

## 3.1 Time course of plasma XOR activity

Plasma XOR activity decreased after PCI and remained low until the discharge of the patients. On the other hand, no significant change was observed in patients undergoing CAG. Fig 1 shows the percentage reduction in pXOR activity at each time point from baseline in patients undergoing CAG and PCI. Compared with CAG group, a significant decrease in the percent pXOR activity from baseline at day 1 and day 2 were observed in the PCI group.

## 3.2 Association of the percentage reduction in plasma XOR activity with the in-hospital outcomes

Characteristics of the PCI procedure and details of the in-hospital outcomes are shown in Table 2. Patients who experienced adverse in-hospital outcomes had a longer total fluoroscopy time and procedure time, and larger volume of the contrast dose (Table 2). The incidence of periprocedural MI was higher in patients undergoing PCI with in-hospital adverse events than in patients without in-hospital adverse events (Table 2). The percentage reduction of pXOR

**Table 1. Baseline characteristics.**

| Characteristics | CAG (n = 25) | PCI (n = 80) | P value |
|---|---|---|---|
| Age, years, mean (SD) | 73.5 (8.4) | 70.2 (10.0) | 0.140 |
| Male, n (%) | 19 (76.0) | 59 (73.8) | 1 |
| BMI, kg/m$^2$, mean (SD) | 24.8 (3.6) | 23.4 (3.9) | 0.110 |
| Hypertension, n (%) | 21 (84.0) | 53 (66.2) | 0.131 |
| DM, n (%) | 14 (56.0) | 37 (46.2) | 0.493 |
| Dyslipidemia, n (%) | 17 (68.0) | 46 (57.5) | 0.484 |
| CKD, n (%) | 17 (68.0) | 43 (53.8) | 0.252 |
| Smoking, n (%) | 1 (4.0) | 10 (12.5) | 0.453 |
| Family history of CAD, n (%) | 0 (0.0) | 3 (3.8) | 1 |
| Hyperuricemia, n (%) | 5 (20.0) | 11 (13.8) | 0.525 |
| Previous MI, n (%) | 12 (48.0) | 33 (41.2) | 0.645 |
| Previous CVD, n (%) | 5 (20.0) | 5 (6.2) | 0.055 |
| Previous PAD, n (%) | 2 (8.0) | 12 (15.0) | 0.51 |
| Previous AF/AFL, n (%) | 2 (8.0) | 10 (12.5) | 0.727 |
| Medications | | | |
| Aspirin, n (%) | 21 (84.0) | 80 (100.0) | 0.003 |
| Clopidogrel, n (%) | 3 (12.0) | 21 (26.2) | 0.178 |
| Prasugrel, n (%) | 10 (40.0) | 58 (72.5) | 0.004 |
| ACE-inhibitor/ARBs, n (%) | 18 (72.0) | 45 (56.2) | 0.242 |
| β blockers, n (%) | 16 (64.0) | 54 (67.5) | 0.81 |
| Statin, n (%) | 22 (88.0) | 72 (90.0) | 0.721 |
| Ca channel blockers, n (%) | 16 (64.0) | 29 (36.2) | 0.02 |
| Diuretics, n (%) | 3 (12.0) | 14 (17.5) | 0.757 |
| XOR inhibitor, n (%) | 3 (12.0) | 11 (13.8) | 0.664 |
| Insulin, n (%) | 4 (16.0) | 10 (12.5) | 0.737 |
| Oral antidiabetic drugs, n (%) | 7 (28.0) | 22 (27.5) | 1 |
| DOAC, n (%) | 2 (8.0) | 8 (10.0) | 1 |
| Warfarin, n (%) | 1 (4.0) | 0 (0.0) | 0.238 |
| Echocardiography | | | |
| LVEF (Teichholz) %, mean (SD) | 60.7 (9.8) | 53.3 (14.9) | 0.155 |
| LVDd, mm, mean (SD) | 49.9 (4.8) | 47.6 (6.4) | 0.355 |
| LVDs, mm, mean (SD) | 33.3 (4.0) | 34.4 (7.3) | 0.703 |
| Laboratory data | | | |
| LDL-C, mg/dL, mean (SD) | 88.8 (23.8) | 73.5 (22.7) | 0.005 |
| HbA1c, %, mean (SD) | 6.9 (0.99) | 6.5 (1.0) | 0.104 |
| Uric acid, mg/dL, mean (SD) | 5.5 (1.7) | 5.6 (1.6) | 0.63 |
| XOR activity, pmol/h/ml plasma, median (IQR) | 31.70 [7.59, 182.00] | 44.65 [0.00, 738.00] | 0.064 |
| Xanthine, μM, median (IQR) | 0.42 [0.25, 22.60] | 0.64 [0.15, 17.30] | 0.14 |
| Hypoxanthine, μM, median (IQR) | 3.58 [0.83, 15.60] | 3.69 [0.70, 63.80] | 0.633 |

Abbreviations: CAG, coronary angiography; PCI, percutaneous coronary intervention; SD, standard deviation; BMI, body mass index; DM, diabetes mellitus; CKD, chronic kidney disease; CAD, coronary artery disease; MI, myocardial infarction; CVD, cerebrovascular disease; PAD, peripheral artery disease; AF/AFL, atrial fibrillation/atrial flutter; ACE, angiotensin-converting enzyme; ARBs, angiotensin II receptor blockers; XOR, xanthine oxidoreductase; DOAC, direct oral anti-coagulants; LVEF, left ventricular ejection fraction; LVDd, left ventricular end diastolic diameter; LVDs, left ventricular end systolic diameter; LDL-C, low density lipoprotein-cholesterol; HbA1C, hemoglobin A1C; IQR, interquartile range.

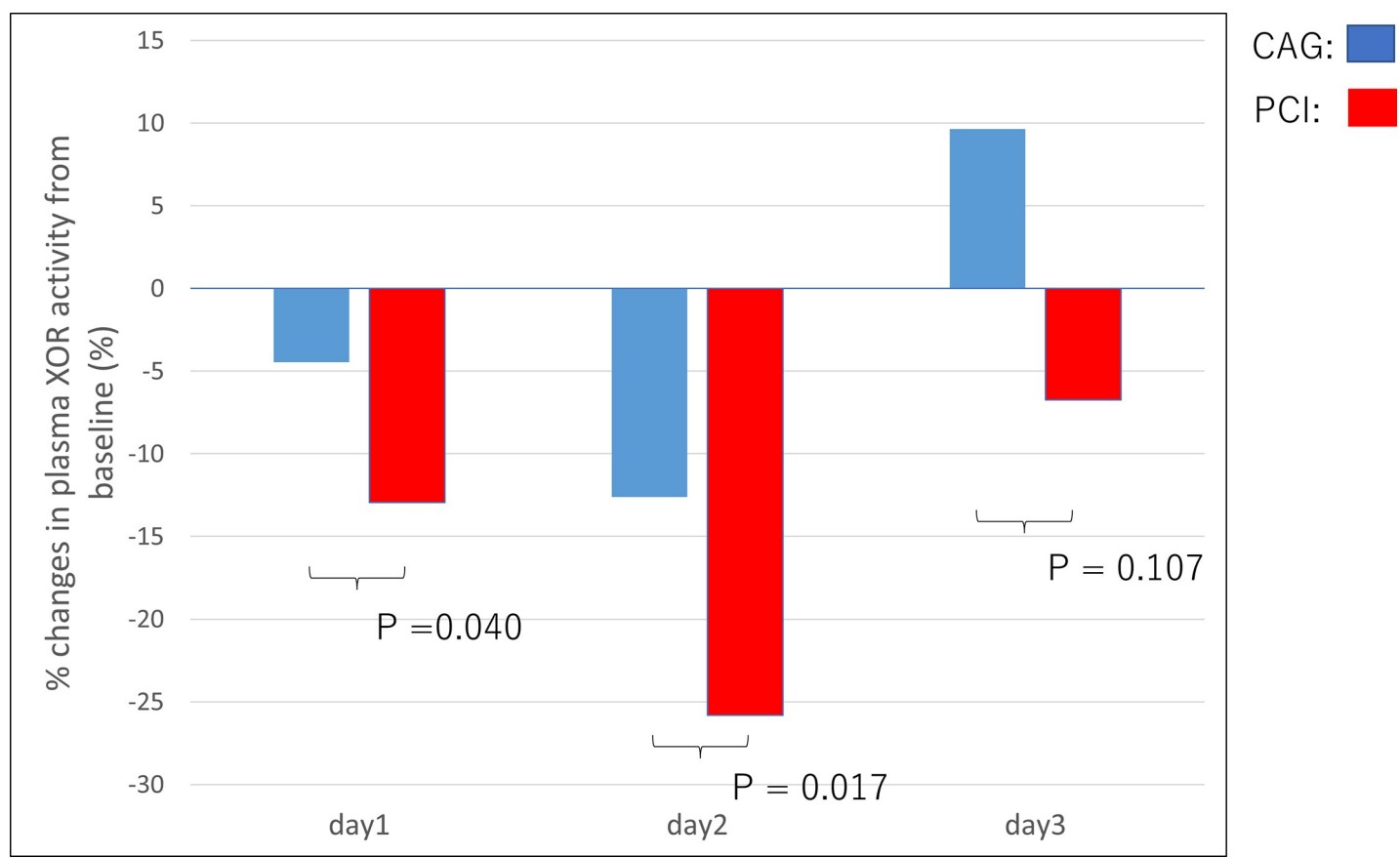

**Fig 1. Comparison of percent changes in plasma XOR activity from baseline at day 1, day 2 and day 3 during the hospitalization between CAG and PCI.**

activity at day 3 in patients undergoing PCI with in-hospital adverse events was higher than in patients without in-hospital adverse events (Fig 2).

### 3.3 Long-term outcomes

The mean duration of follow-up was 11 ± 8 months. During follow-up, a total of 5 patients experienced a non-cardiac adverse event. At 8 months follow-up, there was no significant difference of percentage reduction of pXOR activity at day 3 between patients with and without long-term adverse events (Fig 3).

## 4. Discussion

This study demonstrated that 1) compared with CAG group, a significant decrease in the percent pXOR activity from baseline at day 1 and day 2 were observed in the PCI group, and 2) larger percentage reduction of pXOR activity after PCI from baseline was associated with in-hospital outcomes.

Plasma XOR activity is known to be associated with severity and clinical outcome in chronic heart failure, left ventricular hypertrophy, low left ventricular ejection fraction, coronary artery spasm, and severely decompensated acute heart failure [7, 20–22]. All of these studies indicated an elevated pXOR activity in the patients with cardiac disease suggesting that the severity of these diseases may be attributed to ROS generated by the XOR. However, it is unclear whether invasive procedures used in the treatment of cardiac disease affect pXOR

**Table 2. Details of the PCI procedure and in-hospital outcomes.**

| Characteristics | PCI total (n = 80) | Without AE (n = 72) | With AE (n = 8) | P value |
|---|---|---|---|---|
| Approach site | | | | |
| TRI, n (%) | 72 (90.0) | 65 (90.3) | 7(87.5) | 0.587 |
| TFI, n (%) | 8 (10.0) | 7 (9.7) | 1 (12.5) | |
| Sheath size | | | | |
| 6Fr, n (%) | 69 (86.3) | 63 (87.5) | 6 (75.0) | 0.302 |
| 7Fr, n (%) | 11 (13.7) | 9 (12.5) | 2 (25.0) | |
| Total fluoroscopy time, min, mean (SD) | 34.5 (25.7) | 29.5 (16.5) | 79.88 (46.76) | <0.001 |
| Procedure time, min, mean (SD) | 99.0 (54.5) | 89.2 (41.1) | 188.12 (82.29) | <0.001 |
| Contrast dose, ml, mean (SD) | 127.4 (56.3) | 122.7 (53.3) | 170.00 (71.86) | 0.024 |
| IABP use, n (%) | 1 (1.4) | 1 (1.4) | 0 (0.0) | 1 |
| Rotablator use, n (%) | 7(8.7) | 6 (8.3) | 1 (12.5) | 0.536 |
| Stent | | | | |
| PtCr-EES, n (%) | 14 (17.5) | 11 (15.3) | 3 (37.5) | 0.555 |
| CoCr-EES, n (%) | 37 (46.2) | 33 (45.8) | 4 (50.0) | |
| R-ZES, n (%) | 3 (3.7) | 3 (4.2) | 0 (0.0) | |
| U-SES, n (%) | 22 (27.5) | 21 (29.2) | 1 (12.5) | |
| Others, n (%) | 4 (5.0) | 4 (5.6) | 0 (0.0) | |
| Number of diseased vessels | | | | |
| 1VD, n (%) | 56 (70.0) | 52 (72.2) | 4 (50.0) | 0.287 |
| 2VD, n (%) | 20 (25.0) | 16 (22.2) | 4 (50.0) | |
| 3VD, n (%) | 4 (5.0) | 4 (5.6) | 0 (0.0) | |
| Lesion site | | | | |
| LMT, n (%) | 11 (13.7) | 10 (13.9) | 1 (12.5) | 1 |
| LAD, n (%) | 44 (55.0) | 39 (54.2) | 5 (62.5) | |
| LCX, n (%) | 12 (15.0) | 11 (15.3) | 1 (12.5) | |
| RCA, n (%) | 13 (16.2) | 12 (16.7) | 1 (12.5) | |
| In-hospital outcomes | | | | |
| Deaths, n (%) | 0 (0.0) | 0 (0.0) | 0 (0.0) | 1 |
| MI, n (%) | 6 (7.5) | 0 (0.0) | 6 (75.0) | <0.001 |
| Stroke, n (%) | 0 (0.0) | 0 (0.0) | 0 (0.0) | 1 |
| Shock, n (%) | 1 (1.3) | 0 (0.0) | 1 (12.5) | 0.1 |
| Hemodialysis, n (%) | 0 (0.0) | 0 (0.0) | 0 (0.0) | 1 |
| Blood transfusion, n (%) | 1 (1.3) | 0 (0.0) | 1 (12.5) | 0.1 |
| Major bleeding, n (%) | 1 (1.3) | 0 (0.0) | 1 (12.5) | 0.1 |

Abbreviations: PCI, percutaneous coronary intervention; AE, adverse events; TRI, transradial intervention; TFI, transfemoral intervention; SD, standard deviation; IABP, intra-aortic balloon pump; EES, everolimus-eluting stent; ZES, zotarolimus-eluting stent; SES, sirolimus-eluting stent; VD, vessel disease; LMT, left main trunk; LAD, left anterior descending; LCX, left circumflex; RCA, right coronary artery; MI, myocardial infarction.

activity. This study showed serial changes in pXOR activity in hospitalized patients with stable CAD, undergoing CAG or PCI. The pXOR activity notably decreased after PCI than after CAG, suggesting that PCI may be affecting the pXOR activity by relieving angina and reducing the extent of myocardial ischemia. However, patients with in-hospital adverse events had a higher percentage reduction in pXOR activity than those without in-hospital adverse events. This implies that the influence of a myocardial injury resulting from a longer procedure time is much greater than that of reducing myocardial ischemia, as it produces excessive ROS,

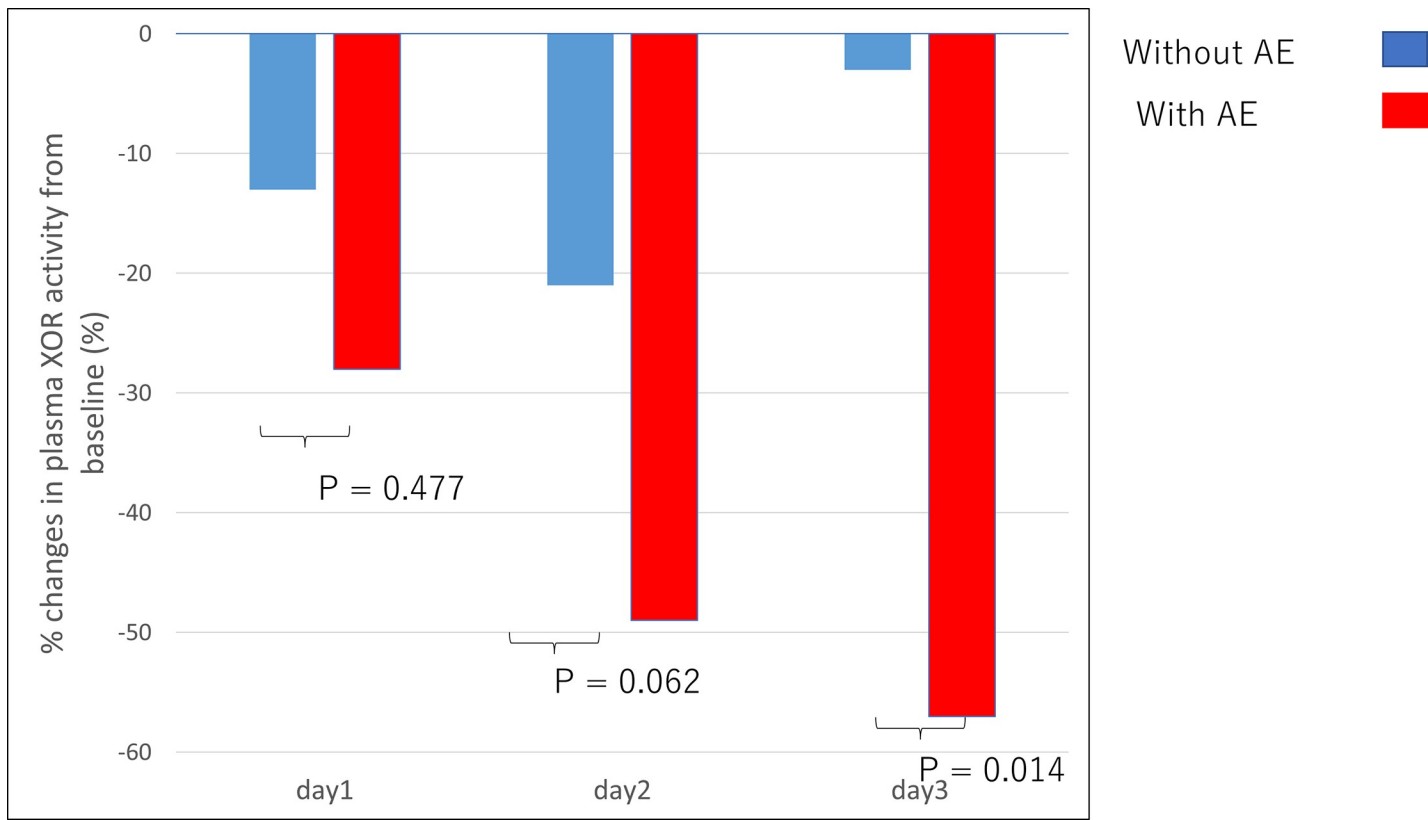

**Fig 2. Percent changes in plasma XOR activity day 1, day 2 and day 3 from baseline in patients with and without composite in-hospital adverse events.**

which in turn inhibits the pXOR activity. To the best of our knowledge, none of the previous studies evaluating the repeated measurements of systemic XOR activity, showed elevated or decreased pXOR activity after PCI in stable CAD patients.

In patients with ST-elevation MI, PCI is the most important strategy used to inhibit the ongoing myocardial damage. Paradoxically, although myocardial reperfusion is essential for myocardial salvage, it comes with a price, it can in itself induce myocardial injury and cardiomyocyte death—a phenomenon termed 'myocardial reperfusion injury' [23]. The reperfused ischemic myocardium generates ROS. These ROS reduce the bioavailability of nitric oxide, hence affects the myocardial damage. There is no evidence, however, of generating ROS by significantly reducing the narrowing of coronary arteries by PCI. In a previous report, the quantity of total peroxides, which is a marker of oxidative stress levels, showed no significant changes during the first 48 h, while their levels declined to below baseline after 30 days post sirolimus-eluting stent implantation [24]. In our study, patients received a drug-eluting stent implantation and pXOR activity instead of the degree of oxidative stress was serially evaluated, since oxidative stress normally occurs when the balance between ROS production and antioxidant defense capacity is disrupted in favor of the former. The exact redox mechanism of CAD is complex and has not yet been fully determined [25]. In fact, the efficacy of pharmacological interventions targeting both the ROS sources and antioxidants in CAD has been inconclusive. Our study may shed some light on the future therapeutic approaches to redox systems in patients with CAD.

Xanthine oxidoreductase is involved in the production of not only ROS, but also uric acid [26]. Several studies have reported the correlation between elevated plasma uric acid levels and

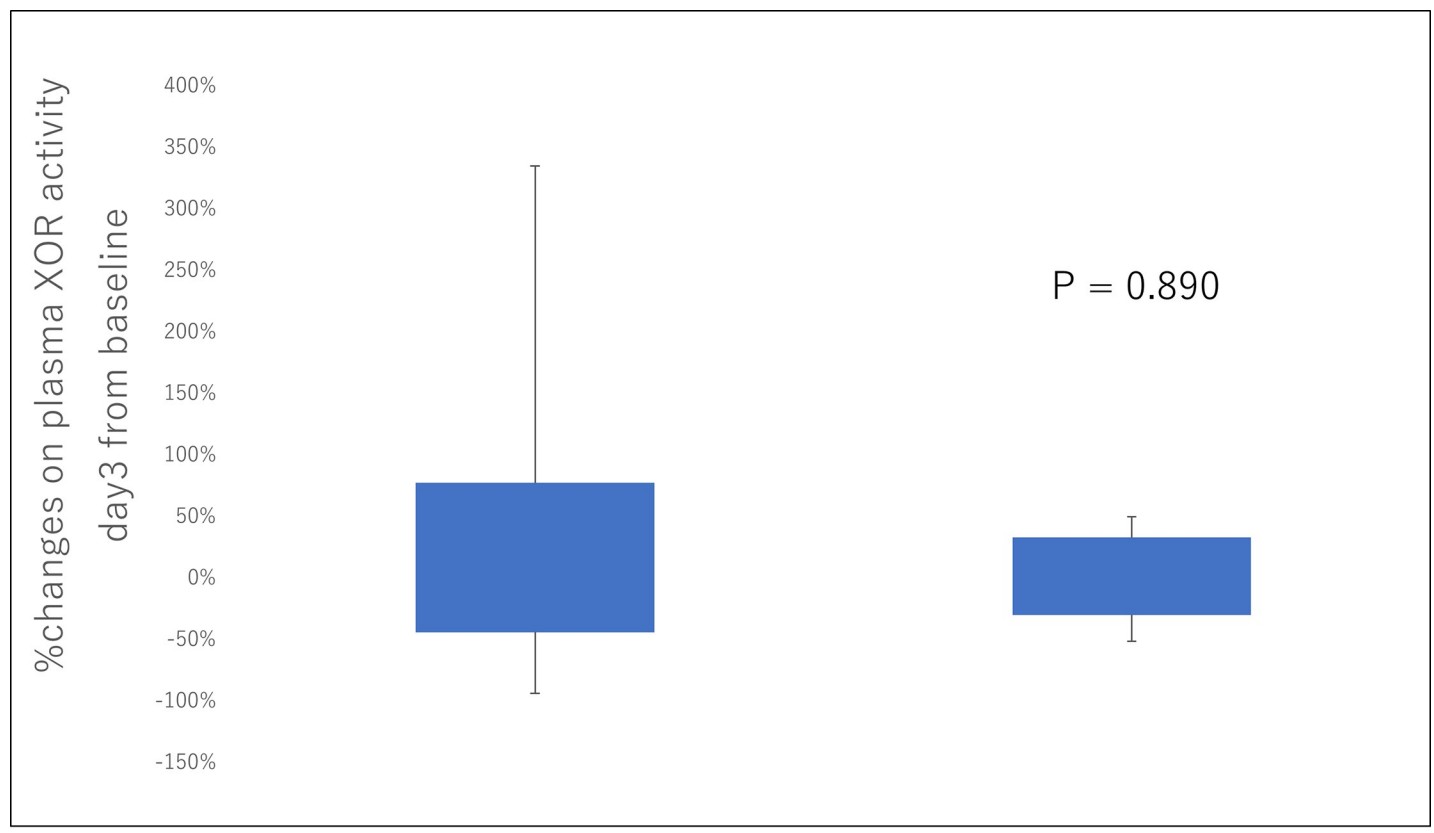

**Fig 3. Percent changes in plasma XOR activity day 3 from baseline in patients with and without long-term adverse events.**

CAD [27, 28]. Meanwhile, low serum uric acid levels were associated with an increased risk of vascular events in men at moderate-to-high cardiovascular disease risk but not in women [29]. Therefore, whether uric acid has a causal relationship in these conditions remains to be determined [30]. The many potential pharmacological cardiovascular benefits of XOR inhibitors include improvement in endothelial function, decrease in tissue oxidative stress, and increase in ATP synthesis in the ischemic tissue [31]. However, the 2019 European Society of Cardiology (ESC) Guidelines for the diagnosis and management of chronic coronary syndromes mentioned that the role of allopurinol, a representative of the XOR inhibitors, in reducing clinical events in CAD remains unclear [32]. We exhibited decreased XOR activity after PCI irrespective of plasma uric acid levels and this may be associated with alleviated myocardial ischemia.

A previous report showed that age, DM, and CKD were associated with a high pXOR activity in outpatients with cardiovascular diseases [33]. Comorbidities including coronary risk factors may influence the baseline pXOR activity in our patients undergoing PCI. That is to say, the background of the patients undergoing PCI was not uniform at the individual level, and therefore, we evaluated the association between the percentage reduction in pXOR activity and in-hospital clinical outcomes instead of assessing the absolute level of pXOR activity. Although the pXOR activity may vary according to patient background, the percent changes in plasma XOR activity may offer superior information regarding the influence of the PCI on pXOR activity.

This study has several limitations. First, the sample size is small although the pXOR activity was measured daily during the hospitalization, implying that several measurements of pXOR activity per patients were required as compared with one-point measurements of pXOR activity. Hence, further investigation is required to clarify the relationship between the change in pXOR activity and in-hospital outcomes. Second, pXOR activity at baseline were higher in patients undergoing PCI than undergoing CAG. Therefore, the percentage reduction of pXOR activity in patients undergoing PCI might be result of a regression to the mean of pXOR activity with alleviated myocardial ischemia by revascularization. Third, to the best of our knowledge, this is the first study to show decreased pXOR activity after PCI in patients with CAD. However, since redox is a complicated process, the mechanism involved in this decrease is unclear, therefore, a verification study should be required to confirm the consistency of the present results.

## 5. Conclusions

In conclusion, patients with stable CAD, showed a significant decrease in the pXOR activity after PCI than after CAG. Furthermore, it was found that a higher percentage reduction of pXOR activity after PCI was associated with in-hospital outcomes.

## Author Contributions

**Conceptualization:** Yuichiro Maekawa.

**Data curation:** Ryota Sato, Keitaro Akita, Takenori Ikoma, Keisuke Iguchi, Yuichiro Maekawa.

**Formal analysis:** Ryota Sato.

**Investigation:** Ryota Sato, Yuichiro Maekawa.

**Methodology:** Ryota Sato, Takayo Murase, Takashi Nakamura, Seigo Akari.

**Resources:** Takayo Murase, Takashi Nakamura, Seigo Akari.

**Supervision:** Yuichiro Maekawa.

**Validation:** Yuichiro Maekawa.

**Writing – original draft:** Ryota Sato, Yuichiro Maekawa.

**Writing – review & editing:** Takayo Murase, Takashi Nakamura, Seigo Akari, Satoshi Mogi, Yoshihisa Naruse, Hayato Ohtani, Yuichiro Maekawa.

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
