## [Decision Letter · Decision Letter 0]

8 Jul 2021

PONE-D-21-00895

Association between plasma xanthine oxidoreductase activity and in-hospital outcomes in patients with stable coronary artery disease after percutaneous coronary intervention

PLOS ONE

Dear Dr. Maekawa,

Thank you for submitting your manuscript to PLOS ONE. After careful consideration, we feel that it has merit but does not fully meet PLOS ONE’s publication criteria as it currently stands. Therefore, we invite you to submit a revised version of the manuscript that addresses the points raised during the review process.

We look forward to receiving your revised manuscript.

Kind regards,

Zhejun Cai, M.D.

Academic Editor

PLOS ONE

Journal Requirements:

1. Please ensure that your manuscript meets PLOS ONE's style requirements, including those for file naming. The PLOS ONE style templates can be found athttps://journals.plos.org/plosone/s/file?id=wjVg/PLOSOne_formatting_sample_main_body.pdf and https://journals.plos.org/plosone/s/file?id=ba62/PLOSOne_formatting_sample_title_authors_affiliations.pdf

3. In your Methods section, please provide additional information about 1) inclusion/exclusion criteria that were applied to participant recruitment,; 2) a priori sample size calculations performed.

'The authors have declared that no competing interests exist.' 

We note that one or more of the authors are employed by a commercial company: name of commercial company.

Additional Editor Comments (if provided):

Reviewers' comments:

Reviewer's Responses to Questions

5. Review Comments to the Author

Reviewer #1: The study by Sato et al. investigates the influence of percutaneous coronary intervention (PCI) on plasma activity of xanthine oxidoreductase (pXOR) They found that PCI was associated with a reduction of pXOR activity. Also, a higher reduction pXOR activity was observed in subjects with peri-procedural adverse events. The article is overall well written and accurate in its analysis, nevertheless, the study has some weaknesses.

- The indications for the revascularization procedure or coronary angiography are not reported in the methods. Were all PCIs elective? Were there differences in indication between the subjects who underwent coronary angiography and those undergoing PCI?

- Since pXOR levels at baseline were higher in patients undergoing PCI, I think it cannot be excluded that the observed reduction in the levels of pXOR is the result of a regression to the mean. This aspect should be discussed at least among the limitations.

- The authors recall that XOR is involved in uric acid production and that high uric acid levels are associated with an increased cardiovascular risk. The authors should mention that even low uric acid levels are independently associated with an increased risk of vascular events as also emerged from a recent study (see J Am Heart Assoc. 2021 Jun;10(11):e020419. doi:10.1161/JAHA.120.020419. Epub 2021 May 17. PMID 33998285).

- The sentence at page 11 “However, the treatment except XOR inhibitor in cardiac diseases influence pXOR activity remains unclear.” Is not clear and should be rephrased.

---

## [Author Response · Author response to Decision Letter 0]

30 Jul 2021

Point-by-point response to the Reviewers’ comments

We thank the editor and the reviewers for their constructive comments. We did our best to improve the manuscript to be acceptable for publication. The responses to all comments are given below; the academic editor’s and reviewers’ comments are in bold font and our replies are in non-bold font.

Journal Requirements:

We have read and understood your journal’s policies, and we believe that neither the manuscript nor the study violates any of these. The manuscript has been rechecked, and the necessary changes have been made in accordance with the reviewers’ suggestions. The changes in the manuscript are highlighted in yellow, and the responses to all comments have been prepared and attached herewith.

I have updated my information and my ORCID iD is now verified.

3. In your Methods section, please provide additional information about 1) inclusion/exclusion criteria that were applied to participant recruitment, 2) a priori sample size calculations performed.

Thank you very much for your comments. We have added inclusion/exclusion criteria in this study for participant recruitment to the Materials and Methods section (Page 5, line 3-Page 6, line 5). In addition, we have added the following reference regarding FFR evaluation to our reference list (Reference #10). Furthermore, we have added a description of the inclusion/exclusion criteria, so we have updated the contents of the study design and added a heading called Participants to describe the inclusion/exclusion criteria in detail (Page 5, line 3-Page 6, line 5). In accordance with the above change, the description on Page 6 line 7 has been changed from coronary angiography to CAG.

Kawaguchi Y, Ito K, Kin h, Shirai Y, Okazaki A, Miyajima K, et al. J Interv Cardiol. 2019 Sep 2;2019:4532862. doi: 10.1155/2019/4532862. eCollection 2019.

Reference #10

We performed a priori sample size calculation and added the description to the statistical analysis section (Page 9, line 15-Page 10, line 1). Furthermore, we have added the following reference regarding sample calculation to our reference list.

Faul F, Erdfelder E, Lang AG, Buchner A. G∗Power 3: a flexible statistical power analysis program for the social, behavioral, and biomedical sciences. Behav Res Methods 2007;39:175–91. Reference #19

'The authors have declared that no competing interests exist.' 

We note that one or more of the authors are employed by a commercial company: name of commercial company.

The funder provided support in the form of salaries for authors [insert relevant initials], but did not have any additional role in the study design, data collection and analysis, decision to publish, or preparation of the manuscript. The specific roles of these authors are articulated in the ‘author contributions’ section.”

First of all, I would like to apologize for the inaccuracy of our funding statement. I would like to provide the amended Funding statement. 

According to the amended Funding statement, I carefully reviewed my statements relating to the authors’ contributions again. It was decided that the author contribution form did not need to be modified and it was left as it was initially written because the funding organization did not play a role in the study design, data collection and analysis, decision to publish, or preparation of the manuscript but only provided financial support in the form of authors' salaries and research materials. That means three of the authors (TM, TN, and SA), who are employed by a commercial company, Sanwa Kagaku Kenkyusyo, are co-investigators, and Sanwa Kagaku Kenkyusyo did not provide any research funds for this study.

5. Please also provide an updated Competing Interests Statement declaring this commercial affiliation along with any other relevant declarations relating to employment, consultancy, patents, products in development, or marketed products, etc. Within your Competing Interests Statement, please confirm that this commercial affiliation does not alter your adherence to all PLOS ONE policies on sharing data and materials by including the following statement: "This does not alter our adherence to PLOS ONE policies on sharing data and materials.” (as detailed online in our guide for authors http://journals.plos.org/plosone/s/competing-interests) . If this adherence statement is not accurate and there are restrictions on sharing of data and/or materials, please state these. Please note that we cannot proceed with consideration of your article until this information has been declared.

Also, I would like to apologize for the inaccuracy of our Competing Interests Statement.

I would like to provide our updated Competing Interests Statement. I have described our updated Competing Interests Statement in the cover letter according to your suggestion.

We have carefully reviewed our reference list and updated it. 

The following articles are not searchable in PubMed because the journals in which they are published are not listed in PubMed.

Nakamura T, Murase T, Satoh E, et al. Establishment of the Process in Blood Sampling and Sample Handling as a Biomarker of Hypoxia-Inducible Diseases; Plasma Hypoxanthine and Xanthine Measurement. Journal of Molecular Biomarkers & Diagnosis 2018; 09(05).

Therefore, the following article has been cited in the revised manuscript in place of the above article.

Furuhashi M, Koyama M, HigashiuraY, Murase T, Nakamura T, Matsumoto M, et al. Differential regulation of hypoxanthine and xanthine by obesity in a general population.

J Diabetes Investig 2020; 11: 878–887. Reference #18

In addition, we have added the following three new references to the reference list in the revised manuscript.

1. Kawaguchi Y, Ito K, Kin h, Shirai Y, Okazaki A, Miyajima K, et al. J Interv Cardiol. 2019 Sep 2;2019:4532862. doi: 10.1155/2019/4532862. eCollection 2019. Reference #10

2. Faul F, Erdfelder E, Lang AG, Buchner A. G∗Power 3: a flexible statistical power analysis program for the social, behavioral, and biomedical sciences. Behav Res Methods 2007;39:175–91. Reference #19

3. Mannarino MR, Pirro M, Gigante B, Savonen K, Kurl S, Giral P, et al. Association Between Uric Acid, Carotid Intima-Media Thickness, and Cardiovascular Events: Prospective Results From the IMPROVE Study. J Am Heart Assoc. 2021 Jun;10(11):e020419. doi:10.1161/JAHA.120.020419. Epub 2021 May 17. PMID 33998285. Reference #29

Reviewer #1: The study by Sato et al. investigates the influence of percutaneous coronary intervention (PCI) on plasma activity of xanthine oxidoreductase (pXOR) They found that PCI was associated with a reduction of pXOR activity. Also, a higher reduction pXOR activity was observed in subjects with peri-procedural adverse events. The article is overall well written and accurate in its analysis, nevertheless, the study has some weaknesses.

1. The indications for the revascularization procedure or coronary angiography are not reported in the methods. Were all PCIs elective? Were there differences in indication between the subjects who underwent coronary angiography and those undergoing PCI?

Thank you for your insightful comments. Yes, all PCI were elective. We have added this point to the exclusion criteria in the Materials and Methods section. This implies that cases of acute coronary syndrome were excluded. Yes, there were differences in indication between the subjects who underwent CAG and those underwent PCI. We have added the indications for the revascularization procedure and coronary angiography to the Materials and Methods section (Page 5, line 3-Page 6, line 5).

2. Since pXOR levels at baseline were higher in patients undergoing PCI, I think it cannot be excluded that the observed reduction in the levels of pXOR is the result of a regression to the mean. This aspect should be discussed at least among the limitations.

Thank you for pointing this out. According to your suggestion, we have added this point to the discussion section (Page 15, line 18-Page 16, line 4). In addition, according to PLOS ONE submission guideline, we have removed the heading “limitations”.

3. The authors recall that XOR is involved in uric acid production and that high uric acid levels are associated with an increased cardiovascular risk. The authors should mention that even low uric acid levels are independently associated with an increased risk of vascular events as also emerged from a recent study (see J Am Heart Assoc. 2021 Jun;10(11):e020419. doi:10.1161/JAHA.120.020419. Epub 2021 May 17. PMID 33998285).

Thank you for your comments. According to your suggestion, we have added the description (highlighted in yellow) about the association between low uric acid levels and an increased risk of vascular events to the discussion section. Furthermore, we have added the suggested reference to the reference list in our revised manuscript.

Meanwhile, low serum uric acid levels were associated with an increased risk of vascular events in men at moderate-to-high cardiovascular disease risk but not in women [29]. Therefore, whether uric acid…(Page 14, line 11-13).

4. The sentence at page 11 “However, the treatment except XOR inhibitor in cardiac diseases influence pXOR activity remains unclear.” Is not clear and should be rephrased.

Thank you for your comment. We have revised the sentence as follows: 

However, it is unclear whether invasive procedures used in the treatment of cardiac disease affect pXOR activity (Page 12, line 15-16).

---

## [Editor Report · Decision Letter 1]

27 Aug 2021

Association between plasma xanthine oxidoreductase activity and in-hospital outcomes in patients with stable coronary artery disease after percutaneous coronary intervention

PONE-D-21-00895R1

Dear Dr. Maekawa,

We’re pleased to inform you that your manuscript has been judged scientifically suitable for publication and will be formally accepted for publication once it meets all outstanding technical requirements.

Kind regards,

Zhejun Cai, M.D.

Academic Editor

PLOS ONE
---

## [Editor Report · Acceptance letter]

6 Sep 2021

PONE-D-21-00895R1 

Association between plasma xanthine oxidoreductase activity and in-hospital outcomes in patients with stable coronary artery disease after percutaneous coronary intervention 

Dear Dr. Maekawa:

I'm pleased to inform you that your manuscript has been deemed suitable for publication in PLOS ONE. Congratulations! Your manuscript is now with our production department. 

Kind regards, 

on behalf of

Dr. Zhejun Cai 

Academic Editor

PLOS ONE